# Exposure to Disinfectants and Cleaning Products and Respiratory Health of Workers and Children in Daycares: The CRESPI Cohort Protocol

**DOI:** 10.3390/ijerph20105903

**Published:** 2023-05-21

**Authors:** Nicole Le Moual, Orianne Dumas, Pierre Bonnet, Anastasie Eworo Nchama, Barbara Le Bot, Etienne Sévin, Isabelle Pin, Valérie Siroux, Corinne Mandin

**Affiliations:** 1Université Paris-Saclay, UVSQ, Univ. Paris-Sud, Inserm, Équipe d’Épidémiologie Respiratoire Intégrative, CESP, 94807 Villejuif, France; 2Scientific and Technical Center for Building (CSTB), Indoor Environment Quality Unit, 77420 Champs-sur-Marne, France; 3Irset (Institut de Recherche en Santé, Environnement et Travail)—UMR_S 1085, Inserm, École des Hautes Etudes en Santé Publique (EHESP), University of Rennes, 35000 Rennes, France; 4EPICONCEPT, 75011 Paris, France; 5Team of Environmental Epidemiology Applied to the Development and Respiratory Health, Institute for Advanced Biosciences, Inserm U 1209, CNRS UMR 5309, Université Grenoble Alpes, 38000 Grenoble, France

**Keywords:** asthma, epidemiology, indoor air quality, settled dust, environmental and occupational exposures

## Abstract

Although cleaning tasks are frequently performed in daycare, no study has focused on exposures in daycares in relation to respiratory health. The CRESPI cohort is an epidemiological study among workers (n~320) and children (n~540) attending daycares. The purpose is to examine the impact of daycare exposures to disinfectants and cleaning products (DCP) on the respiratory health of workers and children. A sample of 108 randomly selected daycares in the region of Paris has been visited to collect settled dust to analyze semi-volatile organic compounds and microbiota, as well as sample indoor air to analyze aldehydes and volatile organic compounds. Innovative tools (smartphone applications) are used to scan DCP barcodes in daycare and inform their use; a database then matches the barcodes with the products’ compositions. At baseline, workers/parents completed a standardized questionnaire, collecting information on DCP used at home, respiratory health, and potential confounders. Follow-up regarding children’s respiratory health (monthly report through a smartphone application and biannual questionnaires) is ongoing until the end of 2023. Associations between DCP exposures and the respiratory health of workers/children will be evaluated. By identifying specific environments or DCP substances associated with the adverse respiratory health of workers and children, this longitudinal study will contribute to the improvement of preventive measures.

## 1. Introduction

Asthma is a chronic respiratory inflammatory disease affecting around 300 million people around the world [1,2,3]. Asthma results from a complex interplay between genetic, environmental, and behavioral risk factors. Clinical expression of the disease varies over the course of life. Asthma appears more frequently in childhood [4] than in adulthood [5]. In children, asthma is the most common chronic disease, affecting 11% of the children in France [4]. Asthma represents a significant economic cost to society [3,6] and is one of the main causes of hospitalizations and absenteeism among children worldwide. There is strong evidence that asthma has its origins in early life, a crucial period in the development of the immune system.

The importance of studying the impact of early life environment on children′s health has been emphasized, especially in the context of the Developmental Origins of Health and Disease (DOHaD) research [7]. Indoor air pollution has been identified as an important risk factor which may induce or exacerbate asthma [8]. Regarding the respiratory health of children, some epidemiological surveys have evaluated the impact of early environmental exposure [1], such as passive smoking, pets, molds, and diet during pregnancy or in early life [9,10], but the role of household exposure to common products such as care products or cleaning products has been scarcely studied [11,12,13]. In industrialized countries, individuals and especially children spend a large part of the day indoors, at home (20 h/day) [14] but also in daycares. Some studies measured indoor air quality in daycares [15], but few have focused on environmental exposures in daycares in relation to children’s respiratory health. Among them, disinfectants and cleaning products (DCP) have a deleterious role in respiratory health [13,16], possibly through damage and permeability of respiratory epithelium [17].

Exposure to DCP is ubiquitous and DCP are commonly used at home [12,13,16,18]. Among adults, household use of these products has been associated with new-asthma onset, current asthma, poor asthma control, and airway inflammation [19,20,21,22,23], consistent with results observed for exposures to DCP at work [24,25,26,27]. In the European Community Respiratory Health Survey (ECRHS), it has been shown that weekly use of such products by women, at work or at home, might induce and accelerate lung function decline comparable to that due to smoking 10 to 20 cigarettes per day over 20 years [28]. Exposure to DCP has also been suggested to be a risk factor for the poor respiratory health of children [12,13]. Among children (0–12 years), an excess risk of respiratory symptoms was suggested due to the daily use of sprays [29] and bleach [30] by parents during home cleaning. Moreover, frequent use of cleaning sprays and daily use of disinfectants during pregnancy were associated with wheezing [31], airway inflammation [32], and asthma [33] in young children.

Exposure to specific DCP in daycares has been scarcely studied [34,35]. However, children who attend daycares may be particularly exposed because cleaning tasks in daycares have been strongly increased over the years [17]. Cleaning tasks are, for hygiene reasons, commonly performed in daycares and often in the presence of children [36]. Among young children, various routes of exposure might be involved: inhalation, dermal contact, and also ingestion due to their habits to often play on the ground and put hands and objects such as toys in their mouths. In the context of the "hygiene hypothesis" [17,37], daycares have been considered a protective environment for the development of asthma and allergies, with the hypothesis of more diverse microbial exposure in children attending daycare. However, while daycare attendance may be associated with more diverse bacterial exposures, studies suggested that infant-care services such as daycares may increase the risk of viral infections [38], which have been associated with increased risk of wheezing and asthma [29].

Despite the increasing number of studies on the impact of exposure to DCP on health, the specific chemical substances involved in impaired respiratory health remain poorly known [16,34], limiting the opportunity to implement appropriate preventive measures. DCP are a complex mixture of irritating (chlorine, ammonia) or sensitizing (limonene) chemicals [12,34,39]. The deleterious role of formaldehyde, a well-known irritant, has been suggested in nocturnal dry cough among children [40]. Moreover, while associations have been suggested between some volatile organic compounds (VOCs) and asthma [41,42], including one study during early childhood [42], associations between specific VOCs emitted by DCP and asthma have been poorly studied [12].

The limited knowledge on the chemical substances involved is partly due to difficulties in evaluating exposure to the multiple cleaning substances in epidemiological surveys [34]. Among available assessment methods, self-reporting is the most commonly used, especially to evaluate exposure to specific products such as bleach or ammonia. However, self-reporting may be a source of classification or memory bias, with potentially differential misclassification bias according to asthma status [43]. The use of the barcodes of products has been proposed to improve exposure assessment regarding product compositions [44]. We have developed an exposure assessment method using a smartphone application to scan DCP barcodes and linked them to product compositions [45,46,47,48].

Respiratory health and diagnostics of asthma are difficult to evaluate in pre-school age children, both in clinics and in epidemiology [49], especially because asthma symptoms are unspecific during early life childhood [50]. Its evaluation is mostly based on the persistence of respiratory symptoms (wheezing and cough), often associated with viral infections [29], but also on the presence of comorbidities such as allergic rhinitis or eczema [51]. In epidemiological studies, the evaluation of respiratory health is often based on standardized questionnaires [52]. Different wheezing phenotypes, characterized by the age of onset, different temporal patterns of symptoms, such as transient early wheezing, late-onset wheezing, or persistent wheezing, have been identified using clinical observations [53] and statistical classification methods [54]. Irritating and nocturnal cough have also been identified as relevant phenotypes in childhood asthma [55]. However, there is no validated objective method for the assessment of young children’s respiratory health that is feasible easily at a large scale [50], while a smartphone application has been proposed to assess rhinitis in adults [56].

Novel major hypotheses to understand the mechanisms underlying the relationship between the early life environment and asthma development include the role of airway microbiota on the one hand, and the role of epigenetic mechanisms on the other hand. The recent development of high-throughput sequencing approaches for human microbiota has provided evidence of its major contribution to health and diseases, such as asthma. Dysbiosis of the respiratory microbiota, potentially involving several bacterial species, such as *Moraxella, Streptococcus,* and *Haemophilus*, has recently been described in children with frequent respiratory infections, wheezing, or asthma [57,58,59,60]. Moreover, several risk factors for asthma in the early years of life, such as number of siblings or daycare setting, have been associated with altered respiratory microbiota in children [57,61,62]. However, the relationship between the daycare environment, including exposure to DCP, the respiratory microbiota, and the respiratory health of children has not been studied. On the other hand, recent studies show that environmental exposures may modify gene expression via epigenetic modifications, especially DNA methylation, and thus influence the health of individuals. For example, epigenetic signatures associated with environmental determinants of asthma, such as living on a farm, smoking, and air pollution have been identified [63,64,65]. To our knowledge, there are no studies specifically addressing the impact of exposure to DCP on DNA methylation.

The aim of the CRESPI study is to examine the impact of environmental exposures to DCP on the respiratory health of workers and children (<4 years) in daycares. Three specific aims were defined: (1) evaluate environmental exposure to DCP by complementary and innovative tools, i.e. indoor air and settled dust measurements in daycares, a specific standardized questionnaire, and identification of substances in the DCP via a database and a smartphone application to scan barcodes and record information related to the product use; (2) evaluate the impact of occupational exposure in daycares on the respiratory health of workers; (3) evaluate the impact of early exposure in daycares on the respiratory health of young children. In addition to the initial aims of CRESPI, three complementary projects are in progress. One aims at further characterizing VOCs and semi-volatile organic compounds (SVOCs) emitted from DCP and care products collected in the daycares, in experimental test chambers. The two other projects will help better understanding of the mechanisms in the relationship between the daycare environment, including DCP exposure, and children’s respiratory health, (i) by studying the associations between the daycare environment, including indoor microbiota, nasal microbiota, and respiratory health in young children attending daycare, and (ii) by evaluating the role of DNA methylation in the relationship between inhaled exposure to chemical substances from DCP and the respiratory health in young children attending daycare.

## 2. Materials and Methods

### 2.1. Study Design

CRESPI (https://crespi.vjf.inserm.fr/; accessed on 16 May 2023; PI: N Le Moual) is a longitudinal epidemiological study including young children attending daycares—aged from 3 months to less than 4 years old—and daycare workers. The CRESPI study aimed to include 100 daycares with detailed characterizations of indoor air (aldehydes and VOCs), settled dust (SVOCs and microbiota), and use of DCP evaluated during a one-day visit, between 2019 and early 2022. The enrollment of around 1000 children (3–45 months at the daycare visit) and 600 adult workers (18–65 years old) was initially targeted. A longitudinal follow-up of children′s respiratory health is planned at least until the end of 2023.

#### Ethical Approval and Consent to Participate

The CRESPI study, coordinated by the French National Institute for Health and Medical Research (Inserm, references C18-05; ID RCB n°2018-A02657-48), has been approved by the French ethic committee ‘Comité de protection des personnes’ (CPP Sud-Est I n°2019-38; May 2019) and the French Data Protection Authority ‘Commission Nationale de l’Informatique et des Libertés” (CNIL n°919185; October 2019). The CRESPI protocol was registered in the clinical trials register (NCT n°04170881). A written informed consent is obtained from all included participants.

### 2.2. Recruitment of Daycares

A random sample, drawn from a national file [36], of 400 daycares in the Paris metropolitan area has been selected. After having tested the protocol in 4 pilot voluntary daycares, we progressively contacted the randomly selected daycares until we reached the aim of at least 100 daycares included. Out of the contacted daycares (n = 185), the acceptance rate was 55% (*n* = 102).

By the end of February 2022, we had reached our goal of visiting 100 randomly selected daycares, with a distribution in terms of geographical location very close to that expected (Table 1).

A total of 108 daycare visits were conducted, including 4 pilot visits and 2 additional non-randomized daycares among volunteers. When a daycare had accepted to participate in the CRESPI study, we mailed them information to be distributed to parents and workers, and we scheduled an in-person meeting with the workers to explain the protocol of the survey. Then, each daycare was visited by fieldworkers from Inserm to: (1) present the objectives of the study to the parents/workers, (2) collect by a smartphone application data on DCP used by workers in the daycare; and by fieldworkers from CSTB to: (3) collect two samples of settled dust via an adapted vacuum cleaner to analyze SVOCs and environmental microbiota, respectively (Figure 1), (4) wipe three different types of surfaces to analyze quaternary ammoniums, (5) sample indoor air for the analyses of aldehydes and VOCs, (6) record ambient parameters (temperature, relative humidity, and carbon dioxide concentration), (7) collect information on the building (surface, ventilation system, flooring material, etc.), its environment (traffic intensity, presence of industries, etc.), and activities of workers including a calendar of cleaning tasks in rooms (toilets, bedroom, etc.) and surfaces (toys, windows, toddler beds, etc.), and (8) sample the main DCP and care products used for the emission tests.

### 2.3. Exposure Assessment

#### 2.3.1. Environmental Measurements

The measurements were performed in the largest room of the daycare in the section of the youngest children during normal occupancy over the day. VOCs were sampled on Tenax 60/80 tubes during 6 h with a Pocket pump (SKC) at an airflow rate of 20 mL/min and then analyzed through gas chromatography coupled with mass spectrometry (GC/MS) according to ISO 16000-6 (2021) standard by the CSTB laboratory. Aldehydes were sampled on DNPH cartridges during 6 h with a Gilair Plus pump (Gilian) at an airflow rate of 300 mL/min and then analyzed through high-performance liquid chromatography coupled with UV detection (HPLC-UV) according to ISO 16000-3 (2011) standard by the CSTB laboratory. The flow rate was checked before and after sampling with a TSI 4146 flowmeter (TSI). VOC samples were sent immediately after sampling at ambient temperature to the laboratory, while aldehyde samples were sent simultaneously in refrigerated packages (4 °C). The target aldehydes were formaldehyde, acetaldehyde, benzaldehyde, hexaldehyde, and nonanal. A total of 66 VOCs were analyzed including benzene, toluene, ethylbenzene, xylenes, styrene, decane, undecane, 2-ethylhexanol, limonene, alpha-pinene, linalool, and siloxanes.

The floor settled dust in the studied room was sampled for SVOC analysis using a vacuum cleaner (Siemens Z 5.0 extreme power edition) modified for the sampling purpose. A cellulose cartridge was inserted into the entrance of the vacuum cleaner tube to collect the dust. The sampling protocol required the vacuum cleaner to be moved across the floor at a speed of 2 m²/min over a total floor area of 10 m² in each room, which corresponded to an average mass of 626 mg (min: 69 mg, median: 432 mg, max: 3628 mg) of sampled dust (n = 106 daycares). The dust samples were sent immediately after sampling in refrigerated packages (4 °C) to the EHESP laboratory. The collected dust in each cartridge was sieved using a Retsch AS 200 vibratory sieve shaker (Retsch GmbH, Haan, Germany) to retain dust with a diameter of ≤100 µm. After sieving, the average mass was 43 mg (min: 1.8 mg, median: 24 mg, max: 705 mg; n = 106 daycares). The sieved dust was then stored at −18 °C in a hermetically sealed 20 mL amber glass flask until chemical analysis by solvent extraction and GC-MS/MS. The target SVOCs were synthetic muscs: tonalide (AHTN) and galaxolide (HHCB); disinfectants / antimicrobials: triclocarban, triclosan and diclosan; isothiazolinones: 2-octyl-2H-isothiazol-3-one (OIT), 1,2-benzisothiazol-3(2H)-one (BIT) and 2-methyl-4-isothiazolin-3-one (MIT); one fungicide: 2-phenylphenol; insecticides: bifenthrin and geraniol; detergents: 4-n-nonylphenol, 4-tert-butylphenol and 4-tert-octylphenol; one UV filter: benzophenone; 4-chloro-3-methylphenol (chlorocresol) used as a disinfectant.

A second settled dust sample was collected with the vacuum cleaner, with the tip previously decontaminated with alcohol (90°) before introducing the sterile Mitest^®^ cartridge. An area (floor covering, floor cushion) of 2 m² was vacuumed. Then, the cartridge containing the collected dust was placed in a 50 mL Falcon tube and put in a zip plastic bag. The samples dedicated to microbiota analysis were stored at room temperature before being sent to INRAE Transfert laboratory (Narbonne, France).

The ambient parameters, i.e., temperature, relative humidity, and carbon dioxide, were measured in the studied room continuously over the sampling day with a 10 min time step with a Class’Air (Pyrescom).

#### 2.3.2. Emissions of VOCs and SVOCs

To complete the missing piece between the products’ composition and the indoor air concentrations of substances emitted by DCP, the emissions of VOCs and SVOCs by the DCP were evaluated in experimental test chambers. Care products used in the studied daycares, such as liquid soap, body wash, and moisturizing cream were also included in the emission tests. The protocol was adapted from the standards relating to building construction products, i.e., ISO 16000-9 (2006) and ISO 16516+A1 (2020) standards, and integrated ad hoc application scenarios for the products to be tested, i.e., spray, liquid product used pure or diluted, or wipe. In all experiments, the test conditions were (i) chamber temperature 23.0 ± 1 °C, (ii) relative humidity 50 ± 5%, and (iii) air change rate 1.5 ± 0.1 h^−1^. The samples were collected before the product application and 30, 60, and 90 min after the application in the test chamber. Aldehydes, VOCs, and SVOCs were sampled and analyzed according to ISO 16000-3 (2011), ISO 16000-6 (2021), and ISO 16516 + A1 (2020) standards.

#### 2.3.3. Smartphone Applications

During a previous collaboration with Epiconcept (http://www.epiconcept.fr/; accessed on 16 May 2023), a barcode smartphone application was developed for the assessment of use of DCP in hospitals [46] and at home [47]. For the CRESPI study, two smartphone applications have been developed with Epiconcept to evaluate (1) exposures—(a) Inserm fieldworkers have scanned barcodes of products used by workers in daycares, and workers have responded to a short face-to-face questionnaire about their use during the visit day, and (b) parents have scanned barcodes of products used at home and responded to a short questionnaire about their use—and (2) respiratory health of children, with monthly follow-up (by the parents).

#### 2.3.4. Database of Cleaning Products

A first database of products scanned in 107 out of the 108 daycares is available and includes 9985 records of products scanned in the 107 daycares for all workers present in the daycare during the visit, corresponding to 891 different commercial products (extract online, Appendix A). In addition, the complete composition (ingredient list) of these 891 products, listed in descending order of ingredient concentration, was recorded from internet searches or after emailing the manufacturer. Currently, the complete composition of 551 products and uncompleted composition for 308 products have been found and are available in a second database. Some products are waiting for a response from the manufacturers (*n* = 5; uncompleted composition mostly found), and 4 manufacturers have refused to provide the exhaustive list of substances in their products (uncompleted composition mostly found). No information was obtained for 32 products. This work regarding the second database is in progress and will be finalized in 2023.

### 2.4. Recruitment and Inclusion of Participants—Sample Size

As of 1 November 2022, out of 108 daycares visited, 2100 workers and 5790 families (parents) were invited to participate in the CRESPI study (Figure 2) through the daycare manager, with three reminders sent to the families by e-mail. Despite difficulties and delay in the recruitment of daycares and participants (Appendix A) due to the COVID-19 pandemic, 1698 parents and workers accepted to participate (reply coupon) and we mailed them questionnaires and consent forms. As of 1 November 2022, 51% (863/1698) returned completed questionnaires and completed/signed consent. Data were collected on a secure platform (Voozanoo) set up by Epiconcept. Automatic email reminders were set up monthly from January 2021 to parents/workers who agreed to participate but did not return their questionnaires/consent. We also set up telephone reminders for participants who have not returned their questionnaires (*n* = 835). Data checking is in progress but the final numbers of participants will be very close to those indicated in Figure 2.

### 2.5. Respiratory Health Assessment

#### 2.5.1. Adults—Daycare Workers

Standardized questionnaires, similar to those used in many studies in Europe including the epidemiological study on genetic and environmental factors in asthma (EGEA; https://cohorte-egea.fr/fr; accessed on 16 May 2023) [66] and the European respiratory health study (ECRHS) [67], were completed at inclusion. For example, we assessed lifetime asthma (having ever had attacks of shortness of breath at rest with wheezing in the chest or asthma attacks during life), and among those with lifetime asthma we defined those with current asthma (having asthma symptoms, an asthma attack, or treatment for asthma in the last 12 months). We also assessed the respiratory health of daycare workers using the validated symptom score for both participants with and without asthma, by the sum of positive responses to the following 5 questions (varies from 0 to 5): wheezing in the chest, waking up with a feeling of breathlessness, breathlessness attack at rest, breathlessness attack after exercise, waking up due to breathlessness in the past 12 months [68]. The questionnaire also includes a question on the age of asthma onset, which will allow us to distinguish adult-onset from childhood-onset asthma. The questionnaire for workers is available in French on the public CRESPI study website (https://crespi.vjf.inserm.fr/wp-content/uploads/2021/04/Questionnaire_recueil_donnees_Personnels_de_creche_V4.0_221020_Final.pdf; accessed on 16 May 2023).

#### 2.5.2. Children

A standardized questionnaire, similar to those previously used in a parent-child cohort (https://cohorte-sepages.fr; accessed on 16 May 2023), collecting information on the use of products, respiratory health, and potential confounders was completed by the parents. For respiratory health, information was collected through the inclusion questionnaire (paper), a smartphone application (monthly), and/or on the participant secured website (monthly or biannual). The questionnaire allows to evaluate respiratory symptoms such as nocturnal cough, bronchiolitis, breathlessness, episodes and frequency of wheezing in the last 12 months, visits to the emergency room or hospitalizations for respiratory symptoms, asthma, and the use of asthma-related treatment, particularly inhaled or oral corticosteroids. We also recorded information on symptoms related to allergic rhinitis and eczema. The questionnaire for children is available in French on the public CRESPI study website (https://crespi.vjf.inserm.fr/wp-content/uploads/2021/04/Questionnaire_recueil_donnees_Enfant_09112020_finale.pdf; accessed on 16 May 2023).

Current wheezing phenotypes at inclusion (2019–2022) will be defined by the inclusion questionnaire, in accordance with definitions used in international studies [28,30,32]: (i) wheezing in the last 12 months (current wheezing); and (ii) recurrent wheezing (at least 3 wheezing episodes in the last 12 months). The severity of wheezing in the last 12 months will be assessed in two ways, based on (i) (a) recurrent wheezing status, (b) discomfort with daily activities, or (c) nocturnal awakening; and (ii) inhaled corticosteroid use and hospitalization (emergency department visit and/or hospitalization due to bronchiolitis, wheezy bronchitis, or asthma attacks) in the last 12 months.

Longitudinal wheezing phenotypes will be evaluated based on the information provided by the parents (i) by questionnaire at inclusion (2019–2022); (ii) every month with the smartphone application and during 1 year after inclusion; and (iii) every 6 months through an online questionnaire (2022 to 2023), from infancy to age 4-years.

Moreover, the respiratory health of the children will be evaluated in 2023 by a group of experts (pediatricians, epidemiologists) using standardized questionnaires and information from medical records available in the children’s health booklets.

### 2.6. Data Collection—Preliminary Results

Various databases are available in the CRESPI study and are illustrated in Appendix A. Data checking and preliminary analyses are in progress.

#### 2.6.1. Characteristics of Cleaning Tasks in the Daycares

The characteristics of cleaning tasks are reported in Table 2.

#### 2.6.2. Characteristics of Daycare Workers

A preliminary descriptive analysis was conducted on the 309 daycare workers whose questionnaires were recorded as of 1 November 2022 (Table 3). The average age of the workers, almost exclusively women, was 43 years, and most of them were never smokers and taking care of the children. Around 15% and one third of the workers have ever had asthma and eczema, respectively.

#### 2.6.3. Characteristics of Children

A preliminary description on the 501 children whose questionnaires were recorded as of 1 November 2022 was performed (Table 4). The average age of the children was 23 months. Approximately 30% of them had wheezing, which is consistent with the prevalence for this age group [29,69].

### 2.7. Biological Assessment

#### 2.7.1. Nasal Swabs for Characterization of Nasal Microbiota

Parents of all children aged ≤ 12 months enrolled in the CRESPI cohort were asked to collect two nasal swabs in their child. The first swab was collected shortly after the child’s daycare visit. The second swab is collected at age 24 months. Swab collection is performed using a standardized protocol tested and validated in the US MARC-35 study [70]. At both collections, we mail the parents a kit to do the swab at home, including collection material (Pediatric FLOQ-Swab [Copan, Brescia, Italy] and a vial containing viral transport media), a specimen transport bag, and a pre-paid mailing box addressed to CSTB. The collection consists of rubbing the swab gently against the inner wall of each nostril and to insert the swab in a provided vial. Detailed instructions are provided to the parents, and a video is available on the study website (https://crespi.vjf.inserm.fr/index.php/ressources/#tutoriels; accessed on 16 May 2023). We ask the parents to place the vial in a specimen transport bag, and to keep it in the refrigerator until they can ship it to CSTB (at their earliest convenience). Samples are stored at CSTB at −80 °C until the end of each collection. After each collection, all samples are shipped on dry ice to Baylor College of Medicine (Houston, TX, USA) for microbiota profiling.

#### 2.7.2. Buccal Cells for Characterization of the DNA Methylation

Parents of children enrolled in the CRESPI cohort were asked to collect buccal swabs in their child using a simple, quick, and non-invasive protocol (via a cytobrush) with a kit specially designed to collect DNA from children by their parents (ORAcollect for Pediatrics/ OC-175, DNAGenotek). A simple step-by-step pictorial description of the procedure was given to parents to correctly perform the collection, which consists of rubbing the inside of each child’s cheek back and forth about 10 times. The sample was sent by the parents to CSTB and stored at room temperature for less than 1 year. DNA was then extracted by a company recognized for this expertise (i.e., DNA Genotek, Qiagen). DNA methylation is measured by a state-of-the-art Illumina chip (EPIC chip) allowing to cover 850,000 CpG sites by the CNRGH (French National Center of Human Genomics research), a laboratory internationally recognized in the genomics and postgenomics field. This is the most widely used method in epidemiological studies.

### 2.8. Statistical Analyses

Statistical analyses will be performed to evaluate the associations between environmental exposures in daycares, evaluated through complementary and precise methods (e.g., measurements of VOCs and SCOVs, DCP emissions, smartphone application), and respiratory health of workers and children, after adjustment for potential confounders. The main outcomes will be current asthma in workers, and wheezing outcomes in children. Among children, longitudinal wheezing phenotypes will be defined by longitudinal latent class analysis (statistical clustering approach), making full use of data collected by questionnaire during follow-up (first monthly and then every six months), from infancy to age 4-years. The impact of DCP exposures on asthma and other respiratory symptoms such as nocturnal cough will be specifically evaluated among adults and children.

For example, among workers, associations of the frequency of use in daycares of cleaning products (such as sprays and wipes) with current asthma and nocturnal cough will be evaluated by logistic regressions, adjusted for age, smoking status, body mass index and educational level. Among children, associations of cleaning products used in daycares with wheezing and nocturnal cough will be evaluated by logistic regressions, adjusted for gender, age, parental smoking status and educational level. Dependency between children from the same daycare will be taken into account by generalized estimating equation models.

Power calculations for exposure variables and respiratory outcomes in children are presented in Table 5. The minimum detectable Odds Ratio for 80% power is close to values observed in previous studies for cleaning exposure at home and respiratory outcomes in children [12].

Analyses will also be conducted to study the relationships between daycare environment, nasal microbiota, and the respiratory health of children. First, we will characterize environmental (dust) and nasal microbiota profiles using clustering approaches, such as partitioning around medoids, using weighted UniFrac distance [59,71]. Then, we will use multinomial logistic regressions: (1) to determine the association of environmental exposures in daycare (number and characteristics of occupants, environmental microbiota (settled dust), indoor air quality, cleaning/disinfection practices) with baseline and longitudinal nasal microbiota profiles; and (2) to determine the association of baseline and longitudinal nasal microbiota profiles with respiratory health (wheezing trajectories from infancy to age 4 years). Finally, we will examine the modulating and/or mediating role of nasal microbiota in the association between environmental exposures in daycare and respiratory health, using stratified analyses and formal test for statistical interaction in multivariable models, and mediation analyses based on the counterfactual framework [72,73].

Analyses will also be conducted to evaluate associations between exposures in daycares and DNA methylation among children. Regarding the epigenetic study, among the multiple exposures to cleaning products measured in CRESPI, we will target in this analysis the exposures associated with respiratory health in first association studies in CRESPI data. An agnostic approach (Epigenome Wide Association Studies, EWAS) based on robust linear regressions adjusted for potential confounders with methylation levels as the response variable and exposure concentration as the predictor will be used. The EWAS results will then be examined by enrichment analysis to determine whether the identified CpG sites are over-represented in specific genomic regions or in specific biological pathways. In addition, association studies simultaneously taking into account several adjacent CpG sites (regions) will be applied to search for differentially methylated regions (DMRs) in relation to exposures to cleaning products.

## 3. Discussion

While early exposure to irritant or sensitizer DCP may have an impact on the respiratory health of young children, few studies have evaluated this topic [12,13]. Among adults, exposure to DCP is a well-known occupational asthma risk factor, but its impact on respiratory health has never been studied among daycare workers [12,18]. The CRESPI cohort has been designed to address these research gaps by collecting detailed exposure data with novel tools and using standardized methods to evaluate respiratory health in ~540 children attending daycares and ~320 daycare workers.

Current knowledge is insufficient to provide recommendations regarding the use of safe DCP, despite a strong demand from both consumers and workers. A recent study suggested that the household use of DCP labelled as ‘green’ might induce fewer risks for current asthma, compared to the use of conventional products [74]. Moreover, an association was observed between weekly household use of DCP classified with a poor Ménag’score^®^ (D and E), a health risk assessment score allowing to inform the consumers by a simple labelling on toxicity of cleaning products ingredients (https://www.60millions-mag.com/2019/08/27/produits-menagers-nocifs-les-premiers-pas-du-menag-score-16406; accessed on 16 May 2023), and more frequent asthma symptoms among 100 women [48]. The use of such a score, with a simple and intuitive labelling, might allow to improve public health prevention. However, additional studies are needed to confirm these recent findings. Moreover, studying DCP as mixtures of complex substances rather than each substance individually is helpful for a better understanding of the health impact of DCP [48]. Data from the CRESPI cohort will allow to further study these recent research topics.

Nevertheless, a weakness of the CRESPI study is that the original purpose of including 1000 children and 600 daycare workers was not reached, linked to the COVID-19 pandemic. A total of 922 parents and 776 workers have responded positively to participate in the study, but only 51% of them have sent their questionnaires and consent as of November 2022. The recruitment and visits of daycares were stopped during the first lockdown and then restarted at a very slow pace (Appendix A). The pandemic might in part explain a low participation rate and return of questionnaire/consent forms from both workers and parents. Participants told us that they did not have time to participate in CRESPI due to the COVID-19 health crisis. Moreover, potential selection bias may have occurred in the inclusion phase for both workers and children. To evaluate selection bias among workers, we will compare the use of DCP among participants and non-participants, using data from the smartphone application and the ingredient list DCP database available for all workers present during the one-day daycare visits. To evaluate selection bias among parents, we will compare information from the reply-coupon of participants who sent their consent and questionnaires to those who did not.

The originality of the CRESPI study lies in the fact that it is the first epidemiological survey focusing on the effect of early environmental exposure to DCP in daycares on respiratory health in the first years of life, as well as the first one which will allow to evaluate associations between occupational daycares exposures and health among workers. Moreover, this cohort was set up through multidisciplinary collaborations between epidemiologists, chemistry researchers, and computer scientists at the cutting edge of technology. The visits of the daycares have been successful; the daycare acceptance rate was higher than initially expected, all workers present the one-day daycare visit accepted to scan DCP used, and all the measurements were conducted as planned. CRESPI benefits from complementary and innovative technologies, via smartphone applications, to improve assessment of daycare exposure to DCP. Indoor pollution (aldehydes and VOCs in indoor air, SVOCs in settled dust) was measured in each daycare. This provides a uniquely precise database characterizing the daycare environment, a common early-life environment for many children. Another strength of the CRESPI study is to collect data on the respiratory health of children and workers through standardized methods (questionnaires) and innovative tools. Moreover, longitudinal data with repeated evaluation of respiratory health will be available in a relatively large population of more than 500 children, which will permit a better understanding of impact of DCP on children′s respiratory health.

## 4. Conclusions

The goal of visiting 100 randomly selected daycares was reached; 9985 products (861 unique ones with barcodes scanned) have been recorded in daycares. A large set of data is available related to these products, with their composition, their VOC, and SVOC emissions, and the indoor concentrations of some of these VOCs and SVOCs, which is unique to date. Inclusion was closed in December 2022 and the follow-up of children is funded until June 2023. Furthermore, we aim to pursue the follow-up of children included in the CRESPI cohort beyond 2023 (conditional to funding) to explore the incidence and evolution of respiratory health of the children to at least the age of 6. Such a cohort, with a rich evaluation of exposure to specific DCP substances and of respiratory health, is essential to evaluate the impact of DCP used in daycares on asthma and to thereafter take adequate preventive measures for both workers and children.

## Figures and Tables

**Figure 1 ijerph-20-05903-f001:**
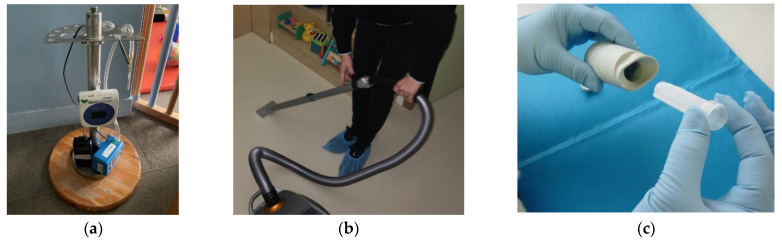
Illustration of some environmental measurements performed in daycares. (**a**) Indoor air sampling for aldehyde and VOC analysis; (**b**) Settled dust sampling with cellulose cartridge for SVOC analysis; (**c**) Settled dust sampling with Mitests^®^ for microbiota analysis.

**Figure 2 ijerph-20-05903-f002:**
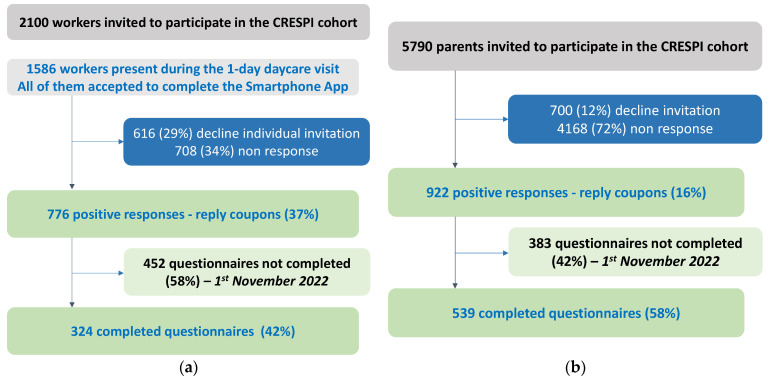
Summary of recruitment—preliminary results. (**a**) Summary of daycare workers recruitment; (**b**) Summary of children recruitment. Around 37% of workers (776/2100) agreed to participate at the first step (**a**). This response rate can be considered as underestimated—as only 1586 workers were at work the day of the visit—and is rather around 49% (776/1586). For the parents, 16% of them (922/5790) have responded positively and 12% (700/5790) have declined to participate. We have not received replies for 72% (4168/5790) of the parents (**b**). However, finally, only ~50% of them returned their questionnaires.

**Table 1 ijerph-20-05903-t001:** Distribution of daycares by administrative department (*n* = 108).

Paris Metropolitan Area	Randomized Daycares (n = 102)			Non-Randomized Daycares (*n* = 6)	
	Expected	Complete Visit	Incomplete Visit *	Pilot Visits	Other Visits
75	34	33	1	2	
92	26	25			
93	17	15	1		1
94	12	13		2	
77	7	7			1
91	4	7			
Total	100	100	2	4	2

* Environmental measurements not performed or cleaning products used by workers not scanned.

**Table 2 ijerph-20-05903-t002:** Characteristics of cleaning tasks in the daycares (*n* = 106 daycares).

Characteristics	*n* (%)
Time for floor wet cleaning (multiple answers)	
in the morning, before children arrive during the day, in presence of children during the day, without children (during the nap) in the evening, after children depart	40 (37.7) 7 (6.6) 11 (10.4) 66 (62.3)
Window opening during floor cleaning	
during cleaning only after cleaning only during and after cleaning never missing information	52 (49.1) 1 (0.9) 31 (29.2) 6 (5.7) 16 (15.1)
Time for furniture cleaning (multiple answers)	
in the morning, before children arrive during the day, in presence of children during the day, without children (during the nap) in the evening, after children depart	29 (27.3) 29 (27.3) 29 (27.3) 72 (67.9)
Window opening during furniture cleaning	
during cleaning only after cleaning only during and after cleaning never missing information	62 (58.5) 3 (2.8) 29 (27.4) 10 (9.4) 2 (1.9)
Number of DCP used in daycares	
1–2 3 4 5 6–9	16 (15.1) 26 (24.5) 30 (28.3) 16 (15.1) 18 (17.0)

**Table 3 ijerph-20-05903-t003:** Characteristics of daycare workers—preliminary results (n = 309 workers).

Characteristics	n (%)
Women	299 (98.4)
Age, years	
Mean ± Standard Deviation [min; max]	43.2 ± 10.2 [18.4; 64.8]
Job	
Children care Cleaner/Cook/Clothes washing Administrative support Others	202 (66.4) 19 (6.3) 46 (15.1) 37 (12.2)
Educational level	
<high school diploma high school to 2-level university >3-level university	98 (32.5) 91 (30.1) 113 (37.4)
Smoking status	
Never smoker Ex-smoker Smoker	194 (64.0) 73 (24.1) 36 (11.9)
Respiratory symptoms	
Asthma symptom score, last 12 months 0 >1 Ever Asthma Asthma attacks, last 12 months Woken by an attack of coughing, last 12 months	261 (85.9) 43 (14.1) 45 (14.8) 21 (7.0) 49 (16.2)
Eczema, ever	105 (34.4)

**Table 4 ijerph-20-05903-t004:** Characteristics of children from daycares—preliminary results (n = 501 children).

Characteristics	*n* (%)
Girls	235 (47.1)
Daycare group	
<1 year old	180 (36.1)
1–2 years old	144 (28.9)
>2 years old	131 (26.3)
Grouped sections	43 (8.7)
Age, months	
Mean ± Standard Deviation	22.7 ± 10.3
Respiratory symptoms	
Wheezing	157 (31.5)
Wheezing and breathlessness	36 (7.2)
Woken by breathlessness	20 (4.2)
Attack of coughing during the night	135 (27.5)
Bronchiolitis	205 (42.6)
Asthma attack	59 (12.1)
Eczema	77 (15.4)

**Table 5 ijerph-20-05903-t005:** Minimum detectable Odds Ratio for 80% power, according to exposure and respiratory outcomes in children (examples for wheezing and nocturnal cough).

	Cleaning, Daycare *	Sprays, Daycare ^†^
Wheezing (31%)	1.51	1.70
Nocturnal cough (27%)	1.56	1.77

Calculation performed with the hypothesis of final population size of 600 children. * For exposure prevalence of cleaning during the day: 46% vs. cleaning tasks performed in the morning before children’s arrival, according to Wei et al. [36]. ^†^ For exposure prevalence of 18%, according to Wei et al. [36].

## Data Availability

Not applicable.

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
