# Peer review of "Exposure to Disinfectants and Cleaning Products and Respiratory Health of Workers and Children in Daycares: The CRESPI Cohort Protocol"

_ijerph, 2023, doi:10.3390/ijerph20105903_

Round 1
Reviewer 1 Report
A manuscript review is normally based on the methods and results and their implications but in this case that is not possible. Providing comments or changes seems pointless because authors will not be able to make any changes as the study is ongoing. Despite that, the following comments are noted:
The study seems well designed and contains a detailed description of methods used in the CRESPI cohort. The methods seem robust and it is understandable the impact of COVID in the recruiting strategy and overall project performance. Still, the recruitment was possible and the results will be very valuable for daycares and parents in general.
Please list the type of quaternary ammonium compounds to be analyzed in wipes.
It would be useful to have a more detailed statistical analysis section.
Author Response
We thank the reviewer for his/her positive comments on our manuscript.
Regarding the type of quaternary ammonium compounds, the target quats are benzethonium chloride, didecyldimethylammonium bromide, benzyldimethyldodecylammonium chloride, benzyldimethyltetradecylammonium chloride, and benzyldimethylhexadecylammonium chloride. However, since the analyses are currently being implemented, the final list of quats to be considered is not yet fixed. In this context, we prefer not to provide the names of the quats in the manuscript.
For the statistical analysis section, we have added the two following sentences “For example, among workers, associations of the frequency of use in daycares of cleaning products (such as sprays and wipes) with current asthma and nocturnal cough will be evaluated by logistic regressions, adjusted for age, smoking status, body mass index and educational level. Among children, associations of cleaning products used in daycares with wheezing and nocturnal cough will be evaluated by logistic regressions, adjusted for gender, age, parental smoking status and educational level. Dependency between children from the same daycare will be taken into account by generalized estimating equations models.” (page 11, line 408-415; revised version).
In addition, for microbiota analyses, we have completed and clarified some sentences of the paragraph (page 11, line 426-429). For the epigenetic analyses, we have added the following sentences: “Regarding the epigenetic study, among the multiple exposures to cleaning products measured in CRESPI, we will target in this analysis the exposures associated with respiratory health in first association studies in CRESPI data. An agnostic approach (Epigenome Wide Association Studies, EWAS) based on robust linear regressions adjusted for potential confounders with methylation levels as the response variable and exposure concentration as the predictor will be used. The EWAS results will then be examined by enrichment analysis to determine whether the identified CpG sites are over-represented in specific genomic regions or in specific biological pathways. In addition, association studies simultaneously taking into account several adjacent CpG sites (regions) will be applied to search for differentially methylated regions (DMRs) in relation to exposures to cleaning products.” (pages 11-12, line 439-449).
Reviewer 2 Report
Overall, a needed objective study of environmental exposures in child care. I would recommend reviewing the materials from the Children's Environmental Health Network's Eco-Healthy Child Care Program, as this is something they have been addressing as well. In particular, https://www.sciencedirect.com/science/article/abs/pii/S0013935115301687?via%3Dihub. (Pg 2, Ln 56)
Specific comments:
Need to define what you mean by "cosmetics"
Who are Inserm?
Pg 6, Ln 260-2: did this already happen or will it occur, need to clarify
Pg 6, Ln 270-4 under database construction: this is confusing
Pg 7, 290-5 under Recruitment: this is also confusing
Pg 11, 409: what is meant by (examples)?
Author Response
We thank the reviewer for his/her careful reading of our manuscript and for his/her pertinent comments/suggestions.
We have completed the sentence and added the suggested reference (page 2, lines 58; revised version).
Specific comments:
Need to define what you mean by "cosmetics": We have replaced ‘cosmetics’ by ‘care products’ in the manuscript and included examples of tested products in the CRESPI cohort (page 6, line 249).
Who are Inserm? : ‘Inserm’ is the French National Institute for Health and Medical Research, and has been defined (page 4, line 165-166).
Pg 6, Ln 260-2: did this already happen or will it occur, need to clarify – this part has been clarified.
Pg 6, Ln 270-4 under database construction: this is confusing - this part has been clarified.
Pg 7, 290-5 under Recruitment: this is also confusing – Few editorial modifications were performed by IJERPH technical team after the submission of our manuscript. In fact, this paragraph should appear as a comment under the Figure 2. We have corrected this point.
Pg 11, 409: what is meant by (examples)? ‘for wheezing and nocturnal cough’ was added in brackets (page 11, 419).
Reviewer 3 Report
Very well-organized and well-written article. Congratulations!
I would only suggest changing the title, which is too long. The adaptation of the keywords to 5, different from the title and the inclusion of the standards with the measurement protocols in the methodology chapter.

Author Response
We thank the reviewer for his/her very positive comments on our manuscript.
"I would only suggest changing the title, which is too long." : We have shortened the title: the final version is ‘Exposure to disinfectants and cleaning products and respiratory health of workers and children in daycares: the CRESPI cohort Protocol’.
"The adaptation of the keywords to 5, different from the title and the inclusion of the standards with the measurement protocols in the methodology chapter.": We have reduced the number of keywords to 5 by removing 4 of them (children, workers, cleaning products, daycares) already included in the title. We have indicated/clarified the standards with the measurement protocols (page 5, line 205-210; revised version).